

# FedLGAN: a method for anomaly detection and repair of hydrological telemetry data based on federated learning

Zheliang Chen[1], Xianhan Ni[1], Huan Li[2] and Xiangjie Kong[2]

[1] Zhejiang Provincial Hydrological Management Center, Hangzhou, Zhejiang, China
[2] College of Computer Science and Technology, Zhejiang University of Technology, Hangzhou, Zhejiang, China

## ABSTRACT

The existing data repair methods primarily focus on addressing missing data issues by utilizing variational autoencoders to learn the underlying distribution and generate content that represents the missing parts, thus achieving data repair. However, this method is only applicable to data missing problems and cannot identify abnormal data. Additionally, as data privacy concerns continue to gain public attention, it poses a challenge to traditional methods. This article proposes a generative adversarial network (GAN) model based on the federated learning framework and a long short-term memory network, namely the FedLGAN model, to achieve anomaly detection and repair of hydrological telemetry data. In this model, the discriminator in the GAN structure is employed for anomaly detection, while the generator is utilized for abnormal data repair. Furthermore, to capture the temporal features of the original data, a bidirectional long short-term memory network with an attention mechanism is embedded into the GAN. The federated learning framework avoids privacy leakage of hydrological telemetry data during the training process. Experimental results based on four real hydrological telemetry devices demonstrate that the FedLGAN model can achieve anomaly detection and repair while preserving privacy.

# INTRODUCTION

With the increasing uncertainty of global natural disasters, the construction of smart hydrology has received more and more attention. Its purpose is to build an integrated hydrological telemetry system that incorporates cloud computing, big data, and other technologies, in order to observe and record hydrological phenomena occurring in nature in a more real-time and accurate manner, providing a data foundation for hydrological research (*Yan et al., 2019*; *Corbari et al., 2019*; *Karimi et al., 2019*; *Kong et al., 2022*). Obviously, as the primary source of hydrological data, hydrological telemetry devices bear the responsibility of data collection and storage. The ability of telemetry devices to

Corresponding author
Xiangjie Kong, xjkong@ieee.org

provide accurate and reliable hydrological data directly affects fundamental decisions such as flood control and drought resistance scheduling, ecological environmental protection, and comprehensive development of water resources. However, in the actual operation process, telemetry devices often encounter issues such as system failures, equipment aging, and weak signals in remote locations, leading to abnormal situations such as numerical errors, partial data loss, and severe data gaps in the collected hydrological data (*Qin & Lou, 2019*). This seriously affects the integrity, authenticity, and accuracy of hydrological data, directly resulting in a significant reduction in the capabilities of various hydrological model statistical analyses. Therefore, identifying anomalies in hydrological data, mining the underlying data features, and simultaneously repairing abnormal data are of great significance for improving hydrological forecasting performance and reducing losses caused by uncertainty in disasters. For time series data such as hydrological telemetry data, existing abnormal detection methods mostly utilize the advantages of long short-term memory (LSTM) networks for learning temporal features and constructing coupled models in combination with other detection algorithms (*Cook, Mısırlı & Fan, 2019*; *Blázquez-García et al., 2021*). *Malhotra et al. (2015)* used stacked LSTM networks to learn higher-level temporal features and made predictions on the data over multiple time steps. Considering the effectiveness and real-time requirements of abnormal detection algorithms, *Ding et al. (2019)* proposed using LSTM models to evaluate the real-time anomalies of each univariate sensor time series, followed by a Gaussian mixture model for multidimensional joint detection of possible anomalies. *Xu et al. (2020)* proposed a new fusion algorithm, LSTM-GAN-XGBOOST, to detect anomalies in deep features of massive time series data. *Niu, Yu & Wu (2020)* introduced an LSTM-based variational autoencoder-generative adversarial network model (LSTM-based VAE-GAN) that jointly trains the encoder and GAN, leveraging the mapping ability of the encoder and the discriminative ability of the discriminator, significantly reducing the time required for anomaly detection. However, the aforementioned studies focus more on anomaly detection rather than repairing the detected abnormal data.

In practical scenarios, the identification and repair of abnormal data often need to be addressed synchronously, which involves detecting the abnormal data and then processing the abnormal portions. Even more, compared to anomaly detection, data repair has greater importance. *Kong et al. (2023)* proposed a dynamic graph convolutional recursive interpolation network (DGCRIN) to interpolate and repair traffic data, which employed a graph generator and dynamic graph convolutional gated recurrent unit (DGCGRU) to perform fine-grained modeling of the dynamic spatiotemporal dependencies of road network. In order to achieve both anomaly detection and repair for time-series data simultaneously, *Zhang et al. (2017)* designed an iterative minimum-change-perception repair algorithm called IMR, which demonstrates high adaptability to existing anomaly detection techniques such as AR and ARX. *Park et al. (2021)* proposed a robust sliding window-based light gradient boosting machine (LightGBM) model, where anomalies are detected using a variational AutoEncoder (VAE), followed by the utilize of random forest to repair the anomalies. Random forest itself is not typically used directly for repair, but rather for anomaly detection and then aiding in the process of deciding how to repair them.

The relevant repair strategies using Random Forest will depend on the nature of anomalies and the domain where the anomalies occur. Typically, it involves various actions such as filling missing values, correcting data errors, removing outliers, and even potentially more complex transformations.

However, the aforementioned studies did not take into account the privacy issues present in the training data. Hence, in this study, we present a GAN model based on the federated learning framework and LSTM network, which performs anomaly detection and reconstructs the time-series data, while ensuring data privacy protection. The model consists of three main components: the federated learning framework, the generative adversarial network model, and the attention-based long short-term memory network. The federated learning framework utilizes its unique mechanism of keeping data local to preserve privacy and employs the federated averaging algorithm to aggregate local training parameters for updating the global model. The generative adversarial network is the core part of the model, composed of a generator and a discriminator, which are optimized through adversarial training. We used the property of the generator in the generative adversarial network to fit real data for data repair. The discriminator's ability to distinguish between real and generated data enables anomaly detection. The attention-based bidirectional long short-term memory network is incorporated to better handle sequential data and further explore the temporal dependencies in hydrological data. Experimental results on four real datasets demonstrate that the GAN model based on federated learning outperforms other control group methods in multiple metrics (training time as well as detection and repair accuracy). This effectively achieves anomaly detection and repair for time series data. The contributions of this article are summarized as follows:

(I)   We propose a distributed model based on the federated learning framework, LSTM and GAN, called FedLGAN, which achieves efficient and accurate anomaly detection and data repair while ensuring data privacy. To the best of our knowledge, it is the first time that the federated learning framework has been used in the context of anomaly detection and data repair for hydrological telemetry data.

(II)   Integrating the attention-based bidirectional LSTM into the generative adversarial network enables effective capturing of the complex dynamics and temporal correlations in hydrological telemetry data. This enhancement strengthens the model's interpretability of anomalies and its capability for data repair.

(III)   By conducting extensive experiments on datasets from four real hydrological stations, we demonstrate the superiority and effectiveness of the proposed FedLGAN model.

The remainder of this article is organized as follows: In the part of related work, we reviewed the cutting-edge methods for anomaly detection and data repair in hydrological telemetry data. Then, we described the foundational knowledge of our framework in the Preliminary section. In the Methodology section, we introduce the proposed framework. The experiment part provides the performance evaluation. Finally, the conclusions are presented.

## RELATED WORK

### Anomaly detection for hydrological telemetry data

Hydrological data anomaly detection is an important research field that holds significant implications for water resource management, flood forecasting, climate change studies, and more. Existing research methods (*Li et al., 2021*; *Chadha et al., 2021*) include statistical modeling approaches such as clustering and classification algorithms, as well as deep learning methods such as convolutional neural networks and recurrent neural networks.

These methods leverage the learning of patterns and features within hydrological data to achieve more accurate detection of anomalies. *Kulanuwat et al. (2021)* developed a median-based statistical outlier detection approach using a sliding window technique. *Shao et al. (2020)* proposed a detection algorithm called AR-iForest, which is a hydrological time series anomaly detection algorithm based on Isolation Forest. It uses an autoregressive model to predict the current data and calculate the confidence interval. Data that falls outside this interval is identified as an anomaly. To enhance the stability of anomaly detection results, *Liu, Lou & Huang (2020)* proposed a parallel anomaly detection algorithm called Flink-iForest, which combines the use of the iForest algorithm with the k-means algorithm to address the threshold partitioning problem. In contrast to their approach, *Sun, Lou & Ye (2017)* proposed a density-based anomaly pattern detection method specifically tailored for large-scale hydrological data with a significant amount of noise. This method addresses the high time complexity issue of traditional anomaly detection algorithms.

Although the above methods demonstrate good performance in detecting extreme and specific value anomalies, they are prone to missing small anomalies. Furthermore, these methods often struggle to uncover the underlying spatiotemporal information in hydrological sequences and fail to provide explanations for the types and causes of anomalies.

### Data repair for hydrological telemetry data

Hydrological telemetry data has always been a scarce and valuable resource. Even more, these data are susceptible to interference, leading to anomalies such as missing values and abrupt changes during the collection and transmission processes. Therefore, the restoration of hydrological data anomalies has always been a research problem of great significance (*Gao et al., 2018*).

The existing hydrological data repair methods primarily involve constructing time series models such as autoregressive (AR) and moving average (MA) models to learn the distribution characteristics of the data. These models are then used to predict, interpolate, or reconstruct the anomalous portions of the data. Among these methods, deep learning-based time series models such as LSTM and RNN are widely applied in practice. *He et al. (2023)* proposed a deep learning model named Con-GRU for repairing water level monitoring data with long-term anomalies, which captures both long-term and local time-dependent features *via* one-dimensional convolution (Conv1D) and gated recurrent units (GRU). *Gill et al. (2007)* proposed a short-term prediction method for groundwater levels in well fields by combining artificial neural networks (ANN) and support vector machines (SVM). They utilized interpolation techniques to fill in missing data and tested their approach

based on the observed data. *Heras & Matovelle (2021)* used automatic learning machines of the Python Scikit Learn module, which integrates a wide range of automated learning algorithms, such as linear regression and random forest.

Currently, there is limited research on the repair of abnormal parts in hydrological data, and previous methods used by researchers have become somewhat outdated and may not be suitable for the current characteristics of multimodal and complex hydrological data. Not only that, but there are also few methods that can simultaneously achieve anomaly detection and data repair. Furthermore, the privacy of hydrological data has been receiving increasing attention, leading to a decrease in available data. Therefore, there is an urgent need for a new method that can ensure data privacy while achieving these two important functionalities.

## PRELIMINARY

### Federated averaging algorithm

The Federated Averaging (FedAvg) algorithm (*McMahan et al., 2017*) describes the process of server-weighted aggregation of local model parameters. In this process, it is assumed that there are $K$ clients in total, and the servers aggregate $t$ times in total. First, the central server initializes the global model $w_t$, and then selects at least one up to k clients to participate in the training. Each selected client simultaneously receives the global model $w_t$ delivered by the server, trains the respective local model $w_{t+1}^k$ with their own data and sends it back to the server. The server will receive all local models and aggregate them in the way of weighted average to get the next round of global model weight $w_{t+1}$, which is calculated by Eq. (1):

$$w_{t+1} \leftarrow \sum_{k=1}^{K} \frac{n_k}{n} w_{t+1}^k. \tag{1}$$

Each participant in the federated learning architecture uses its own local data set to train the model. This involves computing the gradients of the model parameters locally, usually through backpropagation and optimization algorithms such as stochastic gradient descent. During training, participants cannot access other data, thereby achieving the purpose of privacy protection.

### Attention-based bidirectional LSTM

The bidirectional LSTM based on the attention mechanism was initially proposed by *Bahdanau, Cho & Bengio (2014)* for sequence modeling and prediction. It combines the bidirectional LSTM and attention mechanism to better capture contextual information and important features in the input sequence. In the traditional bidirectional LSTM, the input sequence is processed by two LSTM layers in both forward and backward directions. The forward LSTM computes in the order of the input sequence from the beginning to the end, while the backward LSTM computes in the reverse order. In this way, the forward and backward LSTMs capture the forward and backward context information of the input sequence, respectively, generating two sets of hidden state sequences.

To better utilize these hidden state sequences, the attention mechanism is introduced. At each time step of the LSTM, the attention mechanism computes a similarity score between the current hidden state and the input sequence representation *via* dot product, scaled dot product, and concatenation of neural network layers. These scores are then used to compute attention weights for each element in the input sequence, which represent the importance of each input element relative to the current time step. At the same time, a weighted sum of input sequence elements, *i.e.,* context vector, is calculated according to these attention weights. The most relevant information in the input sequence of the current time step of the LSTM can be captublack by the context vector. Finally, the context vector is combined with the output of the LSTM at the current time step as input to the output layer, which generates predictions or further processing. In summary, the attention mechanism allows the model to dynamically assign weights to the inputs based on their importance. It calculates attention weights at each time step, focusing the attention on the most relevant parts of the input sequence for the current prediction. This enables the model to pay more attention to key information in the input sequence, thereby improving the performance of modeling and prediction.

### GAN

The basic idea of GAN (*Goodfellow et al., 2014*) is derived from the "two-player zero-sum game" in game theory, and its main structure contains a generative model $G$ and a discriminative model $D$. Among them, generator $G$ is used to generate data, while discriminator $D$'s main task is to distinguish the real data from the fake data forged by $G \cdot G$ is committed to learning the distribution of real data to fool the discriminator, and the two are optimized in the process of confrontation. The loss function of GAN optimization is as follows:

$$\min_{G}\max_{D}V(D,G) = \quad E_{x\sim P_{data}(x)}[logD(x)] +$$
$$E_{z\sim P_z(z)}[log(1-D(G(z)))]. \tag{2}$$

when $D$ can no longer distinguish the real data from the forged data, the ideal state of GAN training is reached. In our federated anomaly detection task, we use transformer as the generator $G$ of GAN to reconstruct the original sequence, and strengthen the ability of transformer to learn data distribution by means of confrontation.

## METHODOLOGY

In this section, we formally introduce our distributed framework, FedLGAN, which is designed for anomaly detection and data repair. More specifically, it includes the overall framework of FedLGAN, the basic idea behind the framework, and the key technologies.

### Overall framework

The overall framework of FedLGAN is depicted in Fig. 1, which can be divided into three parts: the collaborative training part, the anomaly detection part and the data reconstruction part. The basic idea of FedLGAN is to use the federated learning framework to cooperatively train the data of multiple edge devices, thereby improve the detection
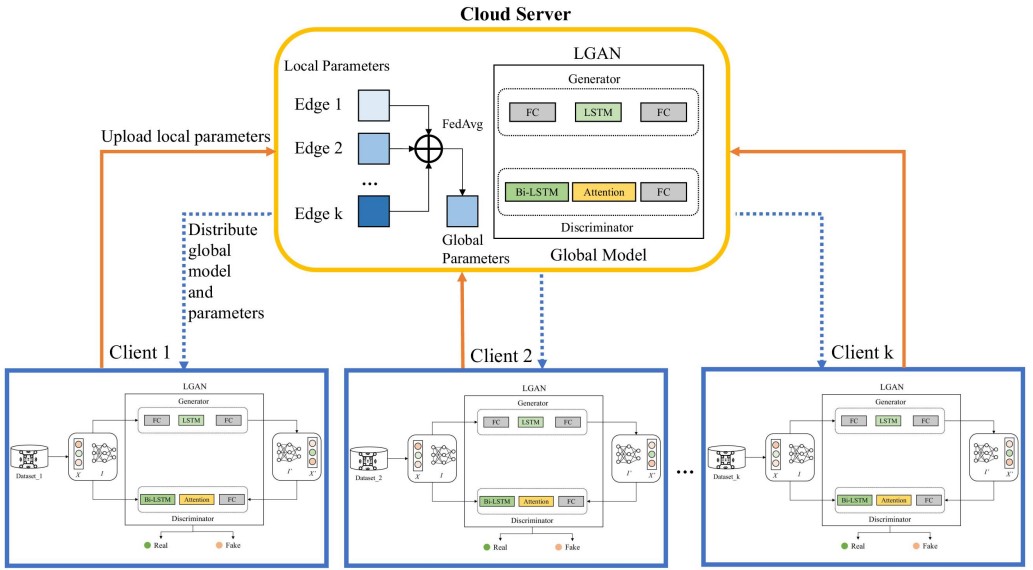

**Figure 1**   Overall framework of FedLGAN.

ability and generalization of the model. Among them, the federated learning framework is used to provide a secure distributed scene to protect the data privacy of edge devices, and the adversarial training mode of GAN is used to enhance the data repair capability of the generator and the anomaly discrimination ability of the discriminator. The LSTM is used to mine the degree of correlation and multi-scale sequence features of sequences. The LSTM is used to improve the GAN's ability to capture the temporal dependencies in time series. We will introduce it in more detail in the following section.

## Collaborative training

Figure 1 shows that the overall structure of the model training is based on the federated learning framework and GAN. In the process of local model collaborative training, we use the cloud server to initialize the global model and distribute it to each edge. After receiving it, the edge will input the local preprocessed normal data into the local model and start training the client. Figure 2 shows the structure of the local model, namely the LGAN.

We further explain the framework of adversarial training stage of FedLGAN in details. As shown in Fig. 2, the generator $G$ and the discriminator $D$ have similar structures, both of which are composed of LSTM blocks. Firstly, we convert the input sequence $X$ into the tensor form $I \in \mathbb{R}^{L \times f}$ with modality, where $L$ represents the length of the sequence, and $f$ is the dimension of potential representation. In the case of Vanilla GAN, neither $G$ nor $D$ have a specific structure to handle time series data. Therefore, the lack of consideration for the unique temporal characteristics of time series data during training is a major reason for the generator's weak ability to fit real data and the discriminator's low accuracy in detecting anomalies. To address these issues, we have introduced LSTM networks and bidirectional LSTM networks with attention mechanism in the $G$ and $D$ parts of the generative adversarial network model, respectively. At the same time, in the training

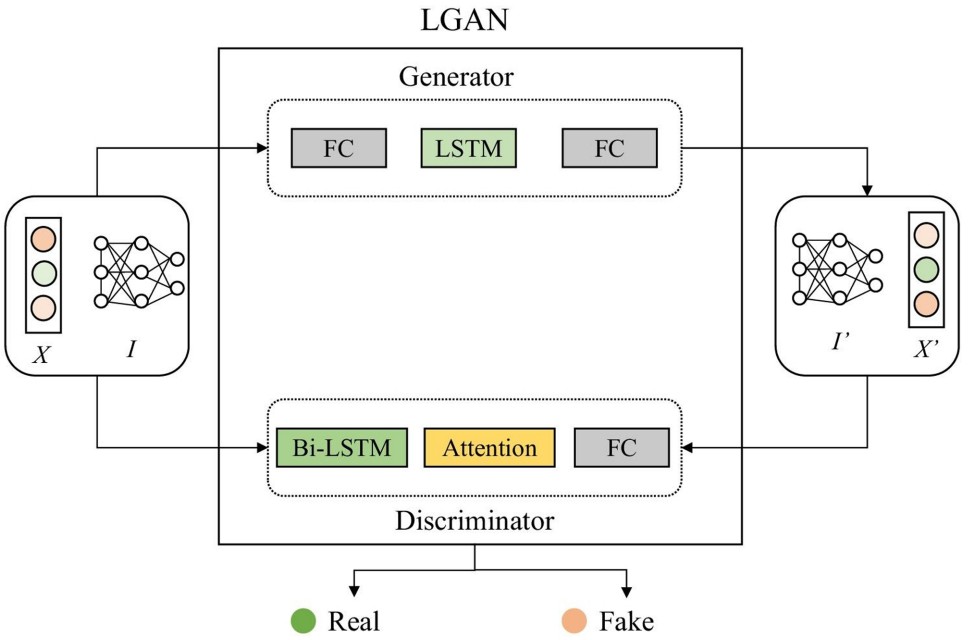

**Figure 2  Structure of LGAN.**

process of GAN, the discrimination ability of the discriminator needs to be slightly greater than the camouflage ability of the generator, otherwise it will easily lead to mode collapse, which refers to a situation where the $G$ produces a limited variety of similar outputs, failing to capture the full diversity of the target distribution.

Therefore, the discriminator is often trained multiple times before the generator is trained once. First initialize the generator $G$ and fix it, start training the discriminator $D$, take the real data $I$ and forged data $I$ as the input of $D$, and pass through the bidirectional LSTM layer, attention layer and The fully connected layer finally outputs the identification result. We feed the processed tensor $I$ into the discriminator $D$. As shown in Figs. 3 and 4, the tensor $I$ first enters the LSTM layer. The core of LSTM is the memory unit, which is cut or added information through a structure called gate to control the circulation and loss of features. This structure determines the degree to which the LSTM unit maintains the previous state and remembers the extracted features of the current data input. It has three gates of the control unit state, which are the input gates, forget gates and output gates are calculated by Eqs. (3), (4) and (5):

$$i_t = \sigma_g(W_i I_t + U_i h_{t-1} + b_i) \tag{3}$$

$$f_t = \sigma_g(W_f I_t + U_f h_{t-1} + b_f) \tag{4}$$

$$o_t = \sigma_g(W_o I_t + U_o h_{t-1} + b_o) \tag{5}$$

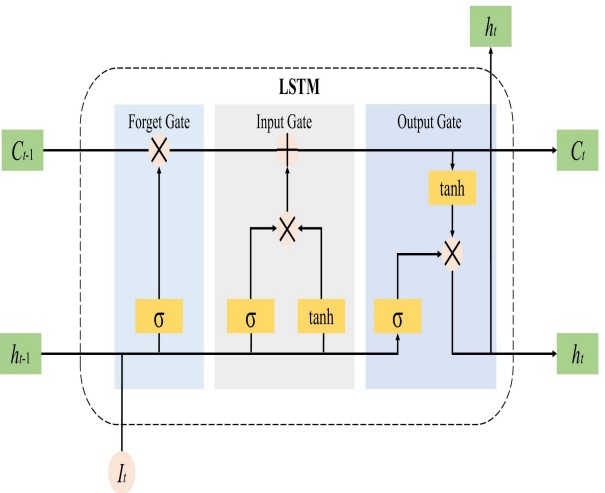

**Figure 3   Internal unit of the LSTM.**

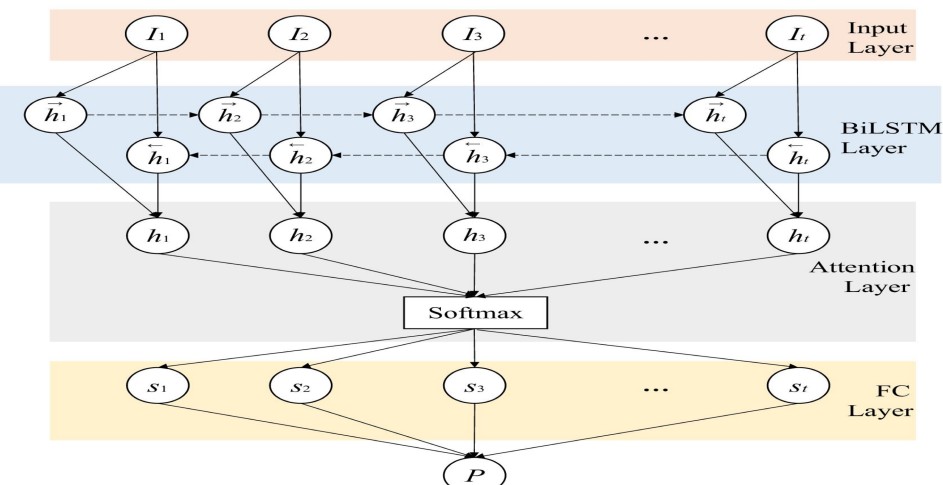

**Figure 4   The details of BiLSTM.**

Then, the cell state at time can be calculated, and the calculation formula is as following equation:

$$C_t = f_t \times C_{t-1} + i_t \times \tilde{C}_t \tag{6}$$

where $i_t$, $f_t$ and $o_t$ are the output values of the input gate, forget gate and output gate at time t respectively, $I_t$ is the t-th input sequence; $h_{t-1}$ refers to the tth time. The hidden layer state of $W$, $U$ and $b$ are the weight vector, parameters and offset of the gating unit, respectively. $\tilde{C}_t$ represents the unit state update value, $\sigma$ is the activation function, and the Sigmoid function is generally used. LSTM is composed of a series of memory unit chains,

and controls the transmission state through gating settings, remembers information that needs to be memorized for a long time, and forgets unimportant content, so as to retrieve the time series variation rules of relatively long intervals and delays in the time series.

The bidirectional LSTM based on the attention mechanism is improved on the traditional LSTM, as shown in Fig. 4, by adding a reverse LSTM layer to the original forward LSTM network layer. The purpose is to consider the context of the two directions to increase the available information of the network (*Schuster & Paliwal, 1997*). Therefore, different from the traditional LSTM, the network structure contains two forward-passed $\vec{h}_t$ and backward-passed $ht^{\leftarrow}$ respectively. The calculation of the hidden layer state $h_t$ at time $t$ is as shown in Eq. (7):

$$h_t = [\vec{h}_t \oplus ht^{\leftarrow}]. \tag{7}$$

To improve the learning ability of the discriminator, an attention mechanism is also introduced. The matrix for extracting the weights of this layer is defined in Eqs. (8) and (9).

$$M = tanh(H) \tag{8}$$

$$\alpha = softmax(w^T M). \tag{9}$$

And use the product of the weight matrix as the output of the attention layer:

$$r = H\alpha^T \tag{10}$$

where $H$ is the output of the LSTM layer, $w^L$ is the transposition of a parameter vector obtained by training and learning, $\alpha$ is the weight matrix, and $r$ is the output of this layer. By adding the improvement of the above structure to the generative confrontation network, the ability of $D$ to detect anomalies and the ability of $G$ to fit the data can be enhanced at the same time, thereby improving the performance of the model as a whole. Finally, we input the result into the fully connected layer network and use the sigmoid activation function to fix the value in the [0,1] interval. Then we can get the probability that each time point $t$ in the time series $I$ is normal, and this probability is defined as $P$. If the input of the discriminator is $I$, that is, real and normal data, the judgment value of the output result at all time points is as close to 0 as possible, otherwise the output tends to 1. Because the discriminator keeps optimizing itself, which means enhancing its detection capability. For normal data, the discriminator is more inclined to correctly recognize it, thus it tends to output 0. Obviously, for the discriminator $D$, whether it is generated data or abnormal data, it is hoped that the output result will be as close to 1 as possible. The training process of $G$ is similar to that of the discriminator $D$. We input the tensor $I$ into $G$, and pass through the LSTM layer and the fully connected layer in turn. After the output can be obtained from Formulas (3), (4), and (5), we can get the reconstructed data $I'$ by adding it to the fully connected layer. In this way, we can reconstruct and replace abnormal data, so as to repair data.

Our local model uses two LSTM blocks, mainly to form an adversarial structure. This way of confrontation forces $G$ to fully learn the characteristic information of normal data, to thereby cheat $D$ in the training process. At the same time, the distinguish transformer $D$ is also trying to distinguish between real data and reconstructed data, which are constantly optimized during the process of confrontation. In the framework of federated learning, $k$ edge devices use their local data for training. After a certain number of iterations, each client uploads its own training parameters to the cloud server for aggregation. Among them, we use the most classic federated average algorithm for aggregation. After that, the cloud server blackistributes the aggregated parameters and models to each client to let them start training again, and so on until convergence. The specific process of model collaborative training is shown in Algorithm 1.

---

**Algorithm 1** Model Collaborative Training Stage

---

**Input:** The generator $G$ and discriminator $D$; the total optimization round $M$; the edge
 devices indexed by $k$ and their training samples $I_k$; initialized global model parame-
 ters: $W_{global}^0$; the local model parameters: $W_{local}^{0,k}$; the ratio of $D$ and $G$ training times
 per round: $N$;

**Output:** a well-trained $G$; a well-trained $D$;

 1:  **for** each round $m = 1, 2, \ldots, M$ **do**
 2:  **for** each edge devices $k$ **do**
 3:   $W_{local}^{m,k} \leftarrow W_{global}^m$
 4:   **for** each round $n = 1, 2, \ldots, N$ **do**
 5:    $\nabla_{\theta_d}[logD(I) + log(1 - D(G(I_k)))]$
 6:   **end for**
 7:   $\nabla_{\theta_g} log(1 - D(G(I_k)))$
 8:   $W_{global}^{m+1} \leftarrow FedAvg[W_{local}^{m,0}, \ldots, W_{local}^{m,k}]$
 9:   $W_{local}^{m+1,k} \leftarrow W_{local}^{m+1}$
 10:  **end for**
 11: **end for**

---

## Optimization method

Since our local model is based on two LSTM blocks, the optimization process of the model meets the training standard of GAN. That is, we update generator $G$ and discriminator $D$ alternately. In the $m$-th iteration, when the $D$ is trained, we fixed $G$ and $D$ is updated according to the following equation:

$$\nabla_{\theta_d} \frac{1}{m} \sum_{m=1}^{M} [logD(I_k^{(m)}) + log(1 - D(G(I_k^{(m)})))] \tag{11}$$

Similarly, we fixed $D$, and updated $G$ according to the following equation:

$$\nabla_{\theta_g} \frac{1}{m} \sum_{m=1}^{M} log(1 - D(G(I_k^{(m)}))) \tag{12}$$

where $M$ denotes the maximum iterations of local training. In order to make the model converge more easily, we also use gradient punishment to force the Lipschitz constraint. Therefore, the loss function of model training stage in the m-th iteration for $D$ can be defined as follows:

$$L_D = \frac{1}{m} \sum_{m=1}^{M} [log D(I_k^{(m)}) + log(1 - D(G(I_k^{(m)})))] \tag{13}$$

$$+ \lambda(||\nabla_{\tilde{I}_k} D(\tilde{I}_k^{(m)})||_2 - 1)^2 \tag{13}$$

where $\tilde{I}_k^{(m)} = \epsilon I_k^{(m)} + (1-\epsilon) G(I_k^{(m)})$ refers the data randomly interpolated and sampled on two vector lines $I_k^{(m)}$ and $G(I_k^{(m)})$. In order to keep the reconstructed data close to the original data, we also take reconstruction loss as the optimization strategy of the generator. As a result, for $G$, we get a new loss function of model training phase in m-th iteration:

$$L_G = \frac{1}{m} \sum_{m=1}^{M} log(1 - D(\hat{I}_k^{(m)})) + ||I_k^{(m)} - \hat{I}_k^{(m)}||^2 \tag{14}$$

## Anomaly detection and data repair

Figure 1 shows the steps of federated anomaly detection and data repair. First, the cloud server distributes the final trained model parameters to all edge clients, and each client updates the parameters in the local model after receiving them. Then, it enters the anomaly detection and data repair stage. In this stage, we input sequence $I$ into generator $G$ and discriminator $D$, and finally we can get the reconstructed sequence $I'$ and the anomaly time points in the detection sequence.

## EXPERIMENT

In this section, we introduce the details of the experiment, including datasets, model settings, evaluation indicators, etc. Then, we compare the performance of our FedLGAN model and other methods. In addition, we also analyzed the hyperparameters of the model.

### Datasets

We used hydrological data collected by four hydrological telemetry devices in Hangzhou, Jinhua, Shaoxing, and Lishui in Zhejiang Province from September 2022 to December 2022 to conduct experiments to ensure that the data sources for model training and testing are authentic and reliable. However, due to the differences in the equipment models and geographical locations of different telemetry sites, the data recording interval and the attributes of the collected data may be different. Therefore, in the experiment, the common attributes of the hydrological equipment of the four telemetry stations are extracted, and the collection records of data such as water level, rainfall, and voltage are counted at intervals of 5 min. In addition, according to the actual situation, the unreasonable abnormal data is divided into a separate test set for the abnormal detection part of the experiment. Considering that the data collected by the device is abnormal only in a few cases, it was necessary to artificially add noise to the normal data to provide a sufficient amount of

abnormal data for testing. Since the data set of each hydrological telemetry equipment contains a total of 90 days of data from January 1 to March 31, 2022, the authors screened all normal data within 90 days as the training set, and the normal and abnormal data of the last 15 days are used as the test set. The experimental data is defined as $X = \{x_1, x_2, \ldots, x_t\}$, where $t$ denotes the number of points in the sequence (the length of the sequence). In addition, considering that the values of different features in multivariate time series data are quite different, we normalize the training data:

$$x_t = \frac{x_t - min(X)}{max(X) - min(X) + \alpha} \tag{15}$$

where $x_t$ represents the vector value at time stamp $t$ in the sequence, $\alpha$ is a small constant to prevent zero-division.

## Experimental settings

All experiments are run on the same server. The host operating system is Ubuntu 18.04, the memory is 128 GB, the CPU is Intel(R) Xeon(R) Gold, 16-core dual-thread, and the graphics card is NVDIA Quadro P6000. The Pytorch version is v1.6.0. We use non-overlapping sliding windows to obtain subsequences. To balance the training efficiency and convergence speed of the model, we set the batch size to 64, and the training process was completed within 50 epochs. At the same time, we use the ADAM optimizer with an initial learning rate of $10^{-4}$. In addition, the experiment uses mean absolute error (MAE), mean square error (MSE), root mean square error (RMSE), and mean absolute percentage error (MAPE) as the evaluation indexes. At the same time, for anomaly detection, we also use commonly used comparison indicators, namely precision, recall and F1 score. Among them, the data of various indicators of anomaly detection are calculated by comparing the detection results of the discriminator in the model with the real labels, while the index data of data repair is obtained by using the formula of normal data before adding noise, against the data repaired by the generator calculated.

## Comparison experiments

In order to reflect the superiority of the generative adversarial network based on federated learning, this experiment compares it with four control algorithms. The comparative algorithms cover parametric methods, non-parametric methods, and deep learning methods. Since this experiment includes both anomaly identification and data repair, it is considered to compare these two parts separately, where anomaly detection part includes LSTM (*Hochreiter & Schmidhuber, 1997*) and GRU (*Cho et al., 2014*), and data repair part includes VAE (*Kingma & Welling, 2013*) and GAN (*Goodfellow et al., 2020*).

- **LSTM (*Hochreiter & Schmidhuber, 1997*)**: a special RNN that performs better on longer sequences.

- **GRU (*Cho et al., 2014*)**: a variant of LSTM that removes the forget gate and consists only of an update gate and a reset gate.

- **VAE (*Kingma & Welling, 2013*)**: a structure composed of an encoder and a decoder, which is trained to minimize the reconstruction error between the encoded and decoded

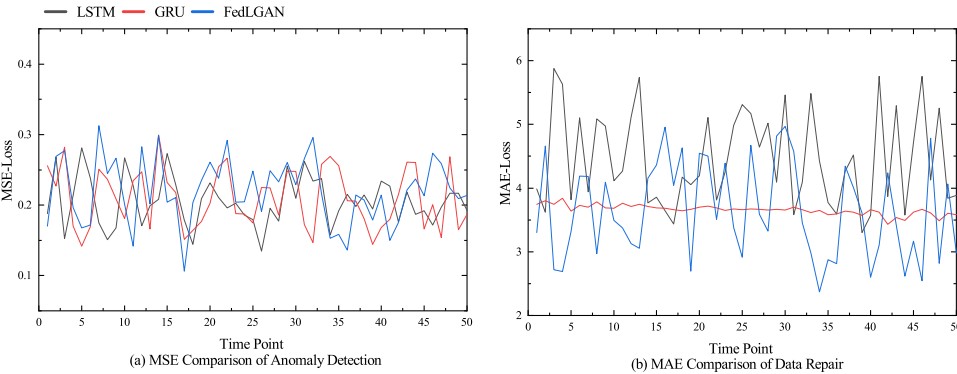

**Figure 5  Performance comparison of different models.**

data and the initial data, and its essence is to estimate the density of a function with hidden variables

- **GAN** (*Goodfellow et al., 2020*): a deep learning model of unsupervised learning, which consists of a generator and a discriminator, and uses the idea of confrontation to continuously optimize the model.

The performance comparison results of different models for anomaly detection and data repair are shown in Figs. 5, and 6 shows the performance of various indicators for model data restoration. More details are shown in Tables 1 and 2, which respectively list the average results of each group's final testing in anomaly detection and data repair. From Table 1, it can be seen that LSTM and GRU have significant advantages in time series prediction compared to the discriminator of GAN model, with various indicator data of 0.371, 0.212, and 13.541%, as well as 0.393, 0.230, and 17.411%, respectively. Although the performance of our model in terms of convergence is not as good as traditional models for processing time series such as LSTM, it still shows superiority in detecting anomalies. We believe that it is mainly the bidirectional LSTM embedded in the GAN that fully learns the potential features of real hydrological telemetry data, thereby being able to distinguish the difference between normal values and abnormal values. From Table 2, it can be seen that GAN, as the backbone of image processing, also performs well in hydrological data restoration work, with various data indicators of 4.420, 1.843, and 85.940%, respectively. However, due to the algorithm not taking into account the temporal characteristics of the data and the potential for pattern collapse, there is still a significant gap between the repaired data and the original normal data. Unlike GAN, VAE explicitly models the distribution of potential variables in hydrological data using encoders and decoders, allowing for the specified distribution of generated data. Therefore, compared to GAN, VAE has a slight performance improvement in hydrological data restoration. Our proposed generative adversarial network model based on federated learning, achieved the optimal experimental results, with three data indicators lower than the control group, which were 3.430%, 1.708%, and 54.824%, respectively.

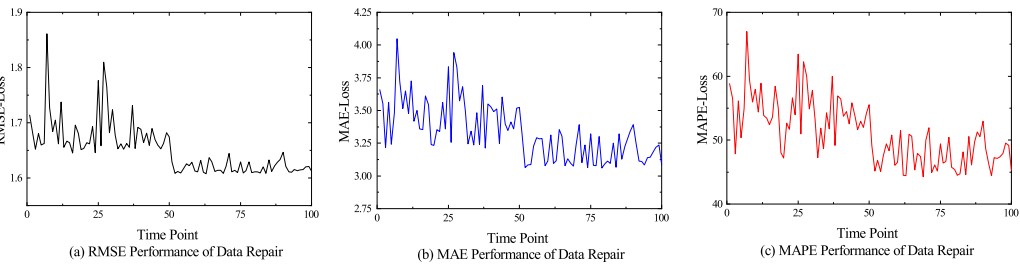

**Figure 6** Performance of data repair.

**Table 1 Comparison of anomaly detection performance.**

| Model | MAE | MSE | MAPE % | Precision | Recall | F1 |
|-------|-----|-----|--------|-----------|--------|-----|
| LSTM | 0.371 | 0.212 | 13.541 | 0.726 | 0.752 | 0.739 |
| GRU | 0.393 | 0.230 | 17.411 | 0.675 | 0.764 | 0.717 |
| FedLGAN | 0.480 | 0.238 | 74.139 | 0.765 | 0.812 | 0.788 |

**Table 2 Comparison of data repair performance.**

| Model | MAE | MSE | MAPE % |
|-------|-----|-----|--------|
| VAE | 3.447 | 2.151 | 72.607 |
| GAN | 4.420 | 1.843 | 85.940 |
| FedLGAN | 3.430 | 1.708 | 54.824 |

## Ablation experiment

In order to demonstrate the effectiveness of the federated learning framework and the attention based long-term and short-term memory network, these two parts were removed and ablation experiments were conducted. The experimental results are shown in Tables 3 and 4.

From Tables 3 and 4, it can be seen that when only using the generative adversarial network model, good experimental results were not achieved in both anomaly detection and data repair. Considering the temporal characteristics of experimental data, FedLGAN* has a MAPE index of 75.037% and 82.639% in anomaly detection and data repair, respectively, which is significantly better compared to the original generative adversarial network model. This change is expected, as LSTM networks can better capture the long-term dependency characteristics of sequence information. In contrast, GAN focuses more on local information and obtains local dependency information between sequences through convolutional neural networks. This change is expected, as LSTM networks can better capture the long-term dependency characteristics of sequence information. In contrast, GAN focuses more on local information and obtains local dependency information between sequences through convolutional neural networks. Unlike FedLGAN, which focuses more on data privacy and security, FedLGAN utilizes a federated learning framework to improve the original model. Its MSE index in anomaly detection is 6.390, while its RMSE index

**Table 3  Performance comparison of anomaly detection ablation experiment.**

| Model | MAE | MSE | MAPE % | Precision | Recall | F1 |
|---|---|---|---|---|---|---|
| GAN | 1.0831 | 1.930 | 92.438 | 0.697 | 0.756 | 0.725 |
| FedLGAN[*] | 0.740 | 0.729 | 75.037 | 0.759 | 0.787 | 0.773 |
| FedLGAN[**] | 2.279 | 6.390 | 91.495 | 0.685 | 0.723 | 0.703 |
| FedLGAN | 0.480 | 0.238 | 74.139 | 0.765 | 0.812 | 0.788 |

**Notes.**
[*]Represents deletion of federated learning framework.
[**]Represents deletion of LSTM.

**Table 4  Performance comparison of data repair ablation experiment.**

| Model | MAE | MSE | MAPE% | Training time/s |
|---|---|---|---|---|
| GAN | 4.442 | 1.843 | 85.940 | 3,868.069 |
| FedLGAN[*] | 4.831 | 2.013 | 82.639 | 3,123.836 |
| FedLGAN[**] | 6.349 | 2.674 | 93.927 | 4,967.078 |
| FedLGAN | 3.430 | 1.708 | 54.824 | 5,334.658 |

**Notes.**
[*]Represents deletion of federated learning framework.
[**]Represents deletion of LSTM.

in data repair is 2.674. Although compared to the performance improvement brought by long-term and short-term memory networks, federated learning frameworks may even have a negative impact on certain indicators , slightly sacrificing the performance of the model in exchange for data privacy and security has significant practical significance and value. It is worth mentioning that due to the unique distributed training of federated learning architecture, it has high requirements for communication, so its model training time is often longer. Considering both data privacy security and its temporal characteristics, all indicators achieved optimal experimental results in the hydrological dataset. Therefore, the introduction of a federated learning framework and a bidirectional long short-term memory network based on attention mechanism in this study have both played a significant role in improving the performance of the model.

## CONCLUSION

We propose a generative adversarial network model based on a federated learning framework, in which the federated learning framework acts on data privacy protection, and the discriminator and generator in the generative adversarial network are used for data anomaly detection and data restoration, respectively. In order to improve the ability of the model to extract temporal features, the two-way long-short-term memory network and the ordinary long-short-term memory network based on the attention mechanism are respectively embedded in the model's discriminator and generator. The model processes the hydrological data of the hydrological telemetry equipment into a time series matrix sequence as input, and extracts relevant time series information from the bidirectional long short-term memory network layer in the discriminator, and uses the result, namely the state of the hidden layer, as the input of the attention layer to obtain weights matrix. Finally, it

outputs the identification result through the fully connected layer to complete the abnormal identification of the data. In addition, the matrix sequence judged as abnormal data by the discriminator is also input to the generator, and its ability to fit the data distribution is used to complete data restoration. The experiments are performed using real hydrological data sets from four telemetry devices in Hangzhou, Jinhua, Shaoxing and Lishui provided by the Zhejiang hydrological communication platform. The results fully prove the feasibility and superiority of the model. However, due to training data are often non-independent and identically distributed, if the data heterogeneity of different selected clients is too large, which may lead to poor performance. Therefore, the generalizability and validity of the model will continue to be verified on the data collected by other hydrological telemetry equipment in different provinces and regions. To be specific, we mainly test and reduce the impact of Non-IID on model performance, such as using data with different degrees of heterogeneity for experimental comparison tests. In addition, considering the distributed training method of the federated learning framework, compared with the centralized model, the operation efficiency is not high. The follow-up work will also focus on reducing the number of communications in federated learning and reducing the training time, so as to further improve the practicability of the network model.

### Funding
This work were supported by the National Natural Science Foundation of China under Grant 62072409, by the Zhejiang Provincial Natural Science Foundation under Grant LR21F020003, and by the R&D Program of of Zhejiang Provincial Department of Water Resources under Grant RB2216. There was no additional external funding received for this study. The funders had no role in study design, data collection and analysis, decision to publish, or preparation of the manuscript.

### Grant Disclosures
The following grant information was disclosed by the authors:
The National Natural Science Foundation of China: 62072409.
The Zhejiang Provincial Natural Science Foundation: LR21F020003.
The R&D Program of of Zhejiang Provincial Department of Water Resources: RB2216.

### Competing Interests
Zheliang Chen and Xiangjie Kong are Academic Editors for PeerJ.

### Author Contributions
- Zheliang Chen conceived and designed the experiments, performed the experiments, analyzed the data, performed the computation work, prepared figures and/or tables, authored or reviewed drafts of the article, and approved the final draft.
- Xianhan Ni conceived and designed the experiments, performed the experiments, analyzed the data, performed the computation work, prepared figures and/or tables, authored or reviewed drafts of the article, and approved the final draft.

- Huan Li conceived and designed the experiments, performed the experiments, analyzed the data, performed the computation work, prepared figures and/or tables, authored or reviewed drafts of the article, and approved the final draft.
- Xiangjie Kong conceived and designed the experiments, performed the experiments, analyzed the data, performed the computation work, prepared figures and/or tables, authored or reviewed drafts of the article, and approved the final draft.

## Data Availability

The partially processed hydrological telemetry data of the four hydrological stations in Zhejiang Province and code are available at GitHub and at Zenodo:

- https://github.com/2450848351/FedLGAN
- Valentine. (2023). 2450848351/FedLGAN: v1.0.0 (v1.0). Zenodo. https://doi.org/10.5281/zenodo.8286185.

## Supplemental Information

Supplemental information for this article can be found online at http://dx.doi.org/10.7717/peerj-cs.1664#supplemental-information.

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
