# Peer review of "FedLGAN: a method for anomaly detection and repair of hydrological telemetry data based on federated learning"

_PeerJ Computer Science, doi:10.7717/peerj-cs.1664_

## Round 0.1 · original submission · Major Revisions

I have received reviews of your manuscript from three scholars who are experts on the cited topic. They find the topic very interesting; however, several concerns must be addressed regarding experimental results, data privacy conservation, English grammar, and comparisons with current approaches. These issues require a major revision. Please refer to the reviewers’ comments listed at the end of this letter, and you will see that they are advising that you revise your manuscript. If you are prepared to undertake the work required, I would be pleased to reconsider my decision. Please submit a list of changes or a rebuttal against each point that is being raised when you submit your revised manuscript.

Thank you for considering PeerJ Computer Science for the publication of your research.

With kind regards,

**Language Note:** The Academic Editor has identified that the English language must be improved. PeerJ can provide language editing services - please contact us at copyediting@peerj.com for pricing (be sure to provide your manuscript number and title). Alternatively, you should make your own arrangements to improve the language quality and provide details in your response letter. – PeerJ Staff

Reviewer 1 ·

Basic reporting

Overall, the paper is well written and maintains a professional and clear use of English throughout. It meets a high standard in terms of clarity and professionalism.

The paper provides ample references to existing literature and sufficient background information, giving appropriate context to the field of study. The structure of the paper is well-organized, adhering to conventional scientific article structure. Figures and tables are clearly labeled and appropriately referenced in the text. However, it would be beneficial if the authors can provide more explicit connection between the figures/tables and the corresponding analysis in the text, which could enhance the reader's comprehension.

The paper appears to be self-contained with relevant results supporting the hypotheses presented. Definitions of terms and theorems are generally clear and detailed. Nevertheless, the paper could benefit from additional explanation or clarification in a few areas:

1. Federated learning: While the concept is mentioned and its benefits discussed, a more in-depth explanation of this method and how it was incorporated into the model would be beneficial. This is especially important as this technique seems to be a key factor differentiating the presented model from previous work.

2. Attention-based LSTM: Similar to federated learning, the explanation of how the attention mechanism was incorporated within the LSTM model could be expanded. Additional details could make it easier for readers unfamiliar with this approach to fully understand the research.

3. Data set and processing: More details on the dataset and how it was preprocessed before being used for training the models would enhance the paper. This would provide readers a better understanding of the context and allow for potential replication of results.

Experimental design

The paper introduces novel research that fits the journal's goals and scope. It presents a meaningful and pertinent research question, which contributes to filling a recognized knowledge gap by employing a federated learning framework and a generative adversarial network model to detect and restore anomalies in hydrological data.

The research seems to uphold a high technical standard, using appropriate machine learning techniques and real-world data. The study also appears to maintain strong ethical standards.

However, there are several areas that need more information:

1. Attention-based LSTM: The paper briefly mentions the usage of an attention-based LSTM but doesn't offer enough detail to fully understand how it was utilized or its contribution to the model's overall performance. More information on this would make it easier to comprehend and replicate the results.

2. Dataset and Preprocessing: The authors note that they used a hydrological dataset from the Zhejiang hydrological communication platform, but they need to provide more information about this dataset, like the number of samples, the type of variables, and any preprocessing steps before it was used for model training. This is essential for the reproducibility of the study.

3. Model Training: More information about the model training process is needed, including details about the optimization algorithms, loss functions, the number of training cycles, batch size, and other model settings.

4. Ablation Study: The authors conducted an ablation study to evaluate the importance of different components in their proposed model. However, the paper would benefit from a more thorough explanation of this process and a more detailed discussion of the results.

Addressing these areas will enhance the paper's overall quality by making the experimental design and findings clearer and more replicable. The authors are urged to revise the manuscript to clarify these points.

Validity of the findings

The findings of this study appear valid and well-supported by the data. The authors have made robust comparisons between their proposed model (FedLGAN) and other established models such as LSTM, GRU, VAE, and GAN, for both anomaly detection and data repair tasks.

Below are some additional points that need to be addressed:

1. Uncertainty Estimates: Uncertainty estimates (such as confidence intervals or standard errors) for the performance metrics would provide a clearer understanding of the variability in the results. This could be particularly important given that the experiments are likely influenced by random factors such as the initialization of model parameters.

2. Model Generalization: While the authors used real-world data from several telemetry devices for their experiments, they acknowledge that performance may not be as strong for other hydrological telemetry equipment due to data limitations. The authors should further discuss how they plan to test and enhance the generalizability of their model.

3. Explanation of the Results: The discussion of the ablation study results should be expanded. The authors should provide an explanation of why removing certain components (like the federated learning framework or LSTM) led to changes in performance. This would provide more insight into the function and importance of each component in the overall model.

Additional comments

I recommend that the paper be accepted after minor revisions.

·

Basic reporting

This article proposes a FedLGAN model to achieve anomaly detection and repair of hydrological telemetry data. Abstract needs to be modified by specifying the results obtained in terms of evaluation parameters and mention how the proposed method is better as compared to the existing techniques. Contributions of the proposed work has been well defined. Figure 1 demonstrating the methodology is not clear. Authors should re generate the clear image.

Experimental design

The description of LSTM and BiLSTM is redundant in the article. Authors need to highlight only the parameter finetuning and additional algorithms applied, if any.

Did authors perform data cleaning/pre-processing on the dataset?

Comparison with existing models for the similar data set has to be analysed in result section. A separate table need to be included.

Conclusion section need to have future enhancement section.

Validity of the findings

This article proposes a FedLGAN model to achieve anomaly detection and repair of hydrological telemetry data. The different models have been applied for anomaly detection and data repair and compared he results in terms of error rate.

The article may be accepted after incorporating the suggestions provided.

Additional comments

NA

Reviewer 3 ·

Basic reporting

Basic reporting:

Grammar (some sentences include more than one correction, please check every comment carefuly):
[lines 13 and 15] data missing -> missing data
[line 18] ...framework and long short term memory network -> ...framework and a long short term memory network...
[line 28] Remove: In recent years,
[line 38] encounter factors -> encounter issues
[lines 46-47] (LSTM) networks in learning temporal features and construct coupled... -> (LSTM) networks for learning temporal features and constructing coupled...
[line 48] (Malhotra...) stacked LSTM -> (Malhotra...) used stacked LSTM
[line 59] Remove: Considering that
[line 61] Even, -> Even more,
[line 61] has more important significance. -> has greater importance.
[line 69] by the use of random forest -> by the use of a random forest
Consider using the word "use" instead of "utilize".
[line 72] we presents -> we present
[line 73] which achieves both anomaly detection and repair for time-series data while... -> which performs anomaly detection and reconstructs the time-series data, while...
[line 76] utilizes its unique mechanism of keeping data local to preserve privacy, and -> uses and unique mechanism that keeps data locally to preserve privacy, and < In this case, since the system works with a client-server architecture, it is not clear to me how data privacy is conserved.
[line 81] for data repair, while the discriminator's -> for data repair. The discriminator's < To keep senteces short.
[line 90] the first time to use the federated learning framework in -> the first time that the federated learning framework has been used in...
[line 98] we will review the cutting-edge -> we reviewd the cutting-edge
[line 99] The, we -> Then, we
[line 100] in the preliminary. In methodology, we -> in the preliminary Section. In the methodology Section, we
[line 101] performance evaluation and concludes the paper finally. -> performance evaluation. Finally, the conclusions are presented.
[line 125] However, these data -> Even more, these data
[line 149] global model.In this -> global model. In this
[line 149] Eqs. 1: -> Eq. 1:
[line 150] Please, rephrase the first sentence, it is not clear.
[line 179] the anomaly detecting part -> the anomaly detection part.
Possibly the term data reconstruction is better than data repairing. Please check this.
[line 192] and start training. -> and start training the client.
[line 198] neither the G nor the D has a specific structure -> neither G nor D have a specific structure
[lines 201-202] The acronym LSTM has already been defined
[line 205] it will easily lead to the problem of mode collapse. -> it will easily lead to mode collapse.
Please, rephrase the following: "According to the above gate function and formula... ...following equation" < It is difficult to read.
[line 211] so as to mine the time series -> so as to retrieve the time series
[After line 212] LSTM network layer, the purpose -> LSTM network layer. The purpose
[After line 212] of the two directions To increase -> of the two directions to increase
[Figure 3] Internal unit of LSTM -> Internal unit of the LSTM.
[After Figure 3] respectively, then the calculation formula of the hidden layer -> respectively. The calculation of the hidden layer
[After Figure 3] at this time is as follows: -> at time t is as shown in Eq. 7 [reference].
[After Eq. 7] the weights of this layer is -> the weights of this layer is defined in Eqs. 8, 9 [reference].
[After Eq. 9] Please rephrase the sentence "And pass the... ...of the attention layer."
[line 215] the ability of the D -> the ability of D
[line 216] the ability of the G -> the ability of G
[lines 217-218] and the use of the activation function sigmoid -> and the use of the sigmoid activation function
[lines 218-219] Please rephrase the sentence: and then we can get... ...I is normal.
[line 222] For the generator G, its training process is -> The training process of G is
[line 226] so as to realize data repair. -> so as to repair data.
[line 229] confrontation forces the G to -> confrontation forces G to
[line 229] normal data, thereby cheat the discriminator D in the training process. -> normal data, to thereby cheat D in the training process.
[line 230] during process of confrontation. -> during the process of confrontation.
[line 231] k should be written using italics.
[After Eq. 12] to force Lipschitz constraint. -> to force the Lipschitz constraint.
[Algorithm 1] Insert period after Model Collaborative Training Stage.
[Algorithm 1] The Generator G and Discriminator D -> The generator G and discriminator D
[Algorithm 1] The words after a ; should not be capitalized. Please correct every entry of the inputs and outpus, e.g. discriminator D; The total -> discriminator D; the total
[After Algorithm 1] the loss function of model training stage in m-th... -> the loss function of the model training stage in the m-th...
[After Algorithm 1] Please rephrase the sentence "to the data that random... ...on the line of I"
[After Algorithm 1] As a result, for the G -> As a result, for G
[line 242] There is an extra space between discriminator D and the comma that follows: discriminator D , and...
[line 242] get the reconstruct sequence -> get the reconstructed sequence
[line 246] Then, compare the performance of our model FedLGAN with other methods. -> Then, we compare the performance of our FedLGAN model and other methods.
[line 249] We used the hydrological -> We used hydrological
[Results Section] Please, use past tense instead of present tense to present your methodology and results. An example of past tense is line 258, where the authors write "it is necessary to artificially...", it is more sound "it was necessary to artificially..." because the experimentation is already finished. The same happens with the Experimental Settings Section.
[line 258] to artificially dirty part of the normal data -> to artificially add noise to the normal data
[line 260] the author screened -> the authors screened
[lines 261-262] Please, rephrase "and the last 15... ...as the test set."
[line 264] the same server, the host -> the same server. The host
[line 266] set to 0.000 1 -> set to 0.0001
[line 269] as the evaluation index. -> as the evaluation indexes.
[line 272] before dirtying and data -> before adding noise, against the data
[lines 278, 279, 280, 282, 284, 287] A blank space should be put between the model's name and the referenced author, e.g. VAE(Kingma and Welling, 2013) -> VAE (Kingma and Welling, 2013)
[Figures 5 and 6] Add periods at the end of each sublegend. E.g. MSE comparison of anomaly detection.
[line 292] The more details -> More details
[line 300] with various indicator data -> with various data indicators
[line 306] on federated learning achieved -> on federated learning, achieved
[line 307] with three indicator data -> with three data indicators
[line 343] Matrix, finally, output the identification -> Matrix. Finally, it outputs the identification
[line 346] The experiment uses real hydrological data -> The experiments are performed using real hydrological data
[line 346] of four telemetry -> from four telemetry


The use of acronyms has many improvement opportunities, consider the following comments:
Use capital letters on the words that are used to define an acronym (please correct all the definitions in the text, this issue appears several times).
E.g. [line 62] proposed a dynamic graph convolutional recursive interpolation network (DGCRIN) -> proposed a Dynamic Graph Convolutional Recursive Interpolation Network (DGCRIN)
[line 68] Please, define the acronym GBM.
In many ocasions, the authors use the full definition of the acronym, though the acronym has already been defined. An example is line 72, where generative adversarial network is written instead of GAN. Please check the whole text and replace full definitions for the acronyms used, the most common issues are with GANs and LSTMs.

References:
[line 107] Which are the papers that use clustering and classification algorithms?
[line 108] Which are the papers that use CNNs and RNNs?
[line 149] What is the reference paper of FedAvg?
References to the Equations are missing in all cases, please include a reference to the paper where the Equations were postulated first. This, unless the equations have been proposed by the authors, in which case it is not necessary to include the reference.
Also, some Equations are not mentioned in the text. For example, for Equation 6 the text reads as following equation:. Please, rather use a legend such as "as shown in Eq. 6." for every equation that is not mentioned in text. A correct example is the way that the authors refered to equations 3, 4, and 5.

Structure and format:
[line 148] This Section is missing an introduction text.
The text of the clients in Figure 1 is very hard to read because it is too small, could the authors increase the font size?
The legend of the figures does not have a congruent format. The authors should revise if they will use capitalized letters at the beggining of each word, or every word in lower case. E.g. Figure 1 uses capital letters and Figure 2 uses lower case.
Please check that every figure legend ends with a period. Missing periods in Figures 1, 2, 3, 4, and Table 2.
Table 1 is missing.
Some lines lack the line number, which made difficult pointing the mistakes. Please, for future revisions try to number every line. For example, there are several paragraphs lacking line numbers after line 205.

Self-content:
[line 61] Could the authors explain why has data reconstruction a greater importance than anomaly detection?
[line 69] How is the random forest used to repair anomalies?
[line 85] The authors should be more specific regarding to which metrics are the ones where the proposed model outperforms others?
[line 94] How does bidirectional LSTM enhance the model's interpretabilty? Possibly the word to be used was comprehension of anomalies, but with interpretability I understand that it is possible to understand more easily how the model reached a decision, if this is the case, could the authors explain why?
[line 149] A brief introduction to federated learning should be provided for the reader.
[line 149] A brief explanation of the client-server architecture used should be explained first.
[line 168] Please provide the reference to the GAN paper.
[line 205] Could the authors very briefly explain what is mode collapse, or at least provide a reference for the reader to consult?
[Text after line 205] Please rephrase all the sentence from Therefore, the discriminator... to ...outputs of the identification result. < It is difficult to understand.
[line 221-222] Please explain why it is hoped that the output of D is as close to 0 as possible.
[line 233] What is the name of the most classic federated average algorithm? (Also, please include the reference).
[line 250] Please, change the expression from the past three months to a more specific period, because when the paper is published, this will no longer stand true.
[Figure 5] There is no analysis in the text as to what this Figure allows one to conclude. Please provide any insight worth mentioning form this figure.
[Figure 6] The label "Performance of data repair." is to broad, could the authors include more information, such as which models are being compared? Even more, this Figure would seem to not be worth displaying at all becuase it is not described and does not allow one to draw any valuable conclusion. The authros should consider omitting this figure.


Results:
[line 25] Could the authors include in the abstract the metcis attained by the model?
The experimental results are not very extensive, since only noise metrics are provided, would it be possible to provide ROC figures or tables with other metrics such as F1-score for anomaly detection?

Experimental design

Experimental design:

Originality:
The main issue that I have with this paper is that the most appealing feature of the proposed model is data privacy conservation. Nevertheless, it is not clear to me how the model conserves privacy. Also, when correctly argumented, this should also be part of the conclusions.
The authors state that other solutions fail to provide explanations for the types and causes of anomalies, but it is not mentioned in the paper how the proposed solution provides this information.

Methods description and replication information:
[line 24] Could the authors explian how does the federated learning framework avoid privacy leakage?
The authors should consider providing the reader with a footnote where the experimentation dataset can be reached (in case that it is going to be public).

Validity of the findings

Validity of the findings:

Impact and novelty clarity:
[lines 120-123] The authors state that other methods miss small anomalies, but in page 11, they state that their model cannot consider all abnormal situations, thus, it does not solve this problem.
[lines 120-123] The authors state that other solutions fail to provide explanations for the types and causes of anomalies, but it is not mentioned int he paper how the proposed solution provides this information.

Conclusion issues:
[line 348] The authors state that their results prove the feasibility and superiority of the model, but they should explicitly point that this is only true for data reconstruction and not for anomaly detection.
[line 348] Could the authors please include any metrics to understand the reach of this study?
[line 352] The authors state that the operation efficiency is not high, but it is no clear to me why this would be truth after reading the paper.

Additional comments

The authors propose a new model called FedLGAN, which is capable of detecting anomalies and reconstructing data. Even more, this model has the potential to conserve data privacy, something that apparently has not been achieved before. The novelty of the model and the features that it provides are appealing for a publication. The results are shown, although some metrics other than only error metrics could be used to compare the model against other proposals. There is a lack of argumentation in on of the most appealing features of the model. This paper should be corrected accordingly and reconsidered for publication.

---

## Round 0.2 · Minor Revisions

All concerns raised by the reviewers have been addressed satisfactorily; however, the paper still needs further proofreading by a fluent English speaker. The manuscript organization, presentation, and discussions on experimental results should also be revised. These issues require a minor revision. If you are prepared to undertake the work required, I would be pleased to reconsider my decision. Please submit a list of changes or a rebuttal against each point that is being raised when you submit your revised manuscript.

**Language Note:** The Academic Editor has identified that the English language must be improved. PeerJ can provide language editing services - please contact us at copyediting@peerj.com for pricing (be sure to provide your manuscript number and title). Alternatively, you should make your own arrangements to improve the language quality and provide details in your response letter. – PeerJ Staff

Reviewer 1 ·

Basic reporting

All of my comments in the previous review have been successfully addressed by the authors in this revision.

Experimental design

All of my comments in the previous review have been successfully addressed by the authors in this revision.

Validity of the findings

All of my comments in the previous review have been successfully addressed by the authors in this revision.

Additional comments

All of my comments in the previous review have been successfully addressed by the authors in this revision. Recommended to accept as is.

·

Basic reporting

Authors have incorporated all the suggestions provided in the revised manuscript. The article may be accepted for the possible publication.

Experimental design

NA

Validity of the findings

NA

Additional comments

NA

Reviewer 3 ·

Basic reporting

The second version presented by the authors is a clear improvement from the first one. I am very thankful with the authors for following my comments and answering to the concerns raised. There are still several issues to solve. My main concern is regarding the results and how the model's performance is not clearly better than any of the other models compared. Also, some of the results are very similar and do not allow one to conclude the superiority of any model. Statistical validation and a clear paragraph explaining (using metrics) how the proposed model is better than the rest is necessary for the paper.

There are other minor issues which I will enlist next. This time, I have provided an attached pdf to allow everyone for a faster and more efficient revision/correction of the paper.

Grammar issues
I have marked several grammar issues in the attached file.
The word blue has been inserted several times, possibly due to the coloring command.
There are still many acronyms that were defined, but the definition is used instead. I have coloured such cases in purple in the attached file.

Format issues
Figures 3 and 4 seem to be stretched, could the authors please verify this?
Could you remark the best results of each metric with bold letters in Tables 1, 2, and 3?
Please, separate the thousands in Table 3 training time using comas. E.g. 3,868 instead of 3868

References
Please, include references to the papers from which the equations are extracted from.

Experimental design

Figure content issues
Figures 5 and 6 should be removed, they do not help to understand the performance of the models, they rather present erratic behaviours.
Please, explain in-line and in the table description of Table 3 what the asterisks mean.
Table 3 is missing results in terms of precision, recall and F1-score.

Validity of the findings

Results issues:
Some of the results, for example the ones presented in Table 1, seem to be very similar. In order to prove the supperiority of a model against another, for those cases where the results are very close, statistical validation should be presented to validate that there is actually an improvement.

Additional comments

Regarding previous revisions:
The authors response: "Random Forest itself is not typically used directly for repair, but rather for anomaly detection and then aiding in the process of deciding how to repair them. The relevant repair strategies using Random Forest will depend on the nature of anomalies and the domain where the anomalies occur. Typically, it involves various actions such as filling missing values, correcting data errors, removing outliers, and even potentially more complex transformations." should be included (briefer) in the paper to explain line 69.

The authors response: "Mode collapse in the context of Generative Adversarial Networks (GANs) refers to a situation where the generator produces a limited variety of similar outputs, failing to capture the full diversity of the target distribution. Instead of generating a wide range of distinct samples, the generator becomes fixated on generating a few dominant modes or patterns. This results in a loss of diversity and richness in the generated data, making the GAN output less representative of the true underlying distribution it's supposed to learn from." should be included in the paper in line 230 for readers unfamiliar with the term.

Authors response "Experimental results on four real datasets demonstrate that the GAN model based on federated learning outperforms other control group methods in multiple metrics (training time, convergence speed, as well as detection and repair accuracy). This effectively achieves anomaly detection and repair for time series data. Please refer to the red or related part of the paper for details." is not reflected on the paper.

1. According to table 3, FedLGAN does not beat other models in time, actually, only time for GAN is provided, training times for LSTM, GRUs and VAEs are missing.

2. There is no convergence speed metric, or at least it is not clear.

3. The FedLGAN obtained the worse results regarding precision, recall and f1-score, could the authors please be more specific in the paper as to how the proposed model achieves better results than the compared models?

Please, add the corresponding answers/comments to the paper.


The authors repsonse: "Because the discriminator keeps optimizing itself, which means enhancing its detection capability. For normal data, the discriminator is more inclined to correctly recognize it, thus it tends to output 0." should be included in the text.

The authors response: "Our model can be applied to both anomaly detection and data repair. Because we train both the discriminator and the generator of the GAN to do both jobs." imply that the proposed model is better than the rest of the compared models. But, according to Table 1, it has the worst results in terms of every metric measured. Could the authors please elaborate on this and explain it too in the paper?

Annotated reviews are not available for download in order to protect the identity of reviewers who chose to remain anonymous.

---

## Round 0.3 · accepted · Accept

I am pleased to inform you that your work has now been accepted for publication in PeerJ Computer Science.

Please be advised that you are not permitted to add or remove authors or references post-acceptance, regardless of the reviewers' request(s).

Thank you for submitting your work to this journal. On behalf of the Editors of PeerJ Computer Science, we look forward to your continued contributions to the Journal.